# Characterization of a Nigerian Lumpy Skin Disease Virus Isolate after Experimental Infection of Cattle

**DOI:** 10.3390/pathogens11010016

**Published:** 2021-12-23

**Authors:** Janika Wolff, Eeva Tuppurainen, Adeyinka Adedeji, Clement Meseko, Olayinka Asala, Jolly Adole, Rebecca Atai, Banenat Dogonyaro, Anja Globig, Donata Hoffmann, Martin Beer, Bernd Hoffmann

**Affiliations:** 1Institute of Diagnostic Virology, Friedrich-Loeffler-Institut, Federal Research Institute for Animal Health, Südufer 10, D-17493 Greifswald-Insel Riems, Germany; Janika.Wolff@fli.de (J.W.); donata.hoffmann@fli.de (D.H.); Martin.Beer@fli.de (M.B.); 2Institute of International Animal Health/One Health, Friedrich-Loeffler-Institut, Federal Research Institute for Animal Health, Südufer 10, D-17493 Greifswald-Insel Riems, Germany; Eeva.Tuppurainen@fli.de (E.T.); anja.globig@fli.de (A.G.); 3Virology Division, National Veterinary Research Institute, Vom 930001, Nigeria; yinkadeji@yahoo.com (A.A.); adolejolly@gmail.com (J.A.); beckybitiyong@gmail.com (R.A.); bbdogonyaro@gmail.com (B.D.); 4Animal Influenza and Other Transboundary Diseases Division, National Veterinary Research Institute, Vom 930001, Nigeria; cameseko@yahoo.com; 5Viral Vaccines Division, National Veterinary Research Institute, Vom 930001, Nigeria; ofcasala@gmail.com

**Keywords:** capripox, lumpy skin disease, LSDV, pathogenesis, Nigeria, virulent, field strain

## Abstract

Lumpy skin disease virus (LSDV), together with sheeppox virus and goatpox virus, belong to the genus *Capripoxvirus* within the family *Poxviridae*. Collectively, they are considered the most serious poxvirus diseases of agricultural livestock. Due to their severe clinical course and consequent loss of production, as well as high mortality of naïve small and large ruminant populations, they are known to have a significant impact on the economy and global trade restrictions of affected countries. Therefore, all capripox diseases are classified as notifiable under the guidelines of the World Organization of Animal Health (OIE). Since the 1970s, several outbreaks of LSD have been recorded in Nigeria. Until now, only a little information on the virus strains leading to the reported outbreaks have been published, dealing mainly with the phylogenetic relationship of those strains and the description of field outbreaks. During the present study, we experimentally infected cattle with a low-passage Nigerian LSDV strain isolated from a skin sample of LSD positive cattle in Nigeria in 2018. Clinical, molecular and serological data indicate that this LSDV isolate is highly pathogenic in cattle since it induced a severe clinical course and approximately 33% mortality in naïve Holstein Friesian cattle after experimental infection.

## 1. Introduction

The genus *Capripoxvirus* consists of lumpy skin disease virus (LSDV), sheeppox virus (SPPV) and goatpox virus (GTPV) [1]. Natural hosts of LSDV are cattle and domestic water buffalo [2,3,4], and transmission is known to occur mainly mechanically via blood-feeding insects [5,6,7] and possibly hard ticks [8]. Furthermore, transmission via shared drinking troughs [9], intrauterine [10] and seminal transmission [11] have been described. It is believed that direct contact between infected and naïve cattle is an inefficient mode of transmission of LSDV [5,12], but actual experimental evidence is still lacking. The severity of clinical signs is highly variable, and the clinical course can range from sub-clinical to acute and prominent signs [2]. The incubation period after experimental infection is 4–14 days [13,14,15,16], and after natural infection is 1–4 weeks [9,17]. Affected cattle display fever [13,14,15], ocular and nasal discharge [15,17], inappetence [12,18] and characteristic pox lesions of the skin that may occur locally restricted or in generalized form [13,14,15,19]. Morbidity and mortality are generally low with an average morbidity rate of 10% [20] and a mortality rate between 1–10% [17,19,21]. Nevertheless, morbidities up to 85% [21,22] and mortalities around 40–75% [17,20] are also reported. In addition to direct losses due to the mortality of affected animals, a drop in milk production [23], reduced weight gain or emaciation [12,18,19], abortions [12,23] and severe damages to hides due to the skin lesions [19,23] have a significant impact on national and global economies. Moreover, control measures are costly, and three consecutive years are required for affected countries to regain their freedom from LSD status [17]. Due to these reasons, capripox virus-induced diseases are classified as notifiable diseases by the World Organization for Animal Health [24].

The control of capripox virus infections is based mainly on the early detection of outbreaks, partial or complete stamping out, quarantine and movement restrictions as well as large-scale vaccination [25,26,27]. Until now, only live attenuated homologous, and heterologous vaccines against capripox viruses are commercially available [18] that were attenuated by multiple passages in cell culture [28] or the chorioallantoic membrane of embryonated chicken eggs [29]. However, these vaccines can induce adverse effects ranging from a local reaction at the inoculation site [12] to generalized skin lesions, the so-called “Neethling-response” [30]. Furthermore, the use of any vaccine against LSDV compromises the disease-free status of the respective country [17,18].

LSDV was first reported in Zambia (sub-Saharan Africa) in 1929 [22,31]. In August 1989, the first outbreak of LSDV was reported outside of Africa in Israel [32], and since 1990, outbreaks in the Middle East have been reported [17]. However, until 2012, only sporadic outbreaks in the Middle East were recorded [18]. In 2015, LSDV was first reported in the European Union and had spread through different Balkan countries [33]. By the end of 2017, control measures were successful, and the spread of LSDV was stopped [27]. In Nigeria, LSDV was first reported in 1974 and 1976, followed by reports in different regions of the country. Nomadic herds and organized farms were affected [34,35,36,37]. Initial molecular characterization of LSDV strains circulating in Nigeria between 2000 and 2016 revealed that there was a change in LSDV strains in Nigeria. A strain isolated in 2000 was found to be more similar to the Kenyan NI-2940 strain, whereas strains isolated between 2010 and 2016 showed higher genetic similarity to strains that were present in European countries during that time [38].

The aim of our work was the molecular and virological analysis of a current LSDV field strain from Nigeria using European cattle and the associated unambiguous pathogenetic characterization of the viral strain. During the present study, cattle were experimentally infected with an LSDV isolate obtained from Nigeria (LSDV-V/281-Nigeria) in 2018, and clinical signs, viremia, viral shedding and seroconversion were analyzed. This Nigerian LSDV isolate proved highly virulent in cattle, and systemic signs could be observed in some of the affected animals.

## 2. Results

### 2.1. Clinical Signs

The first clinical sign observed after experimental infection was increased body temperature, which started at 5 dpi with individual animal R-41 exhibiting 39.7 °C. In the following days, the body temperature of R-81 (39.6 °C, 6 dpi), R-84 (39.8 °C, 7 dpi) and R-83 (39.8 °C, 8 dpi) started to increase. In the case of R-41 and R-83, the increased body temperature lasted only for three and two days, respectively. Fever was first detected at 7 dpi in animal R-81 (40.5 °C), followed by R-74 (40.6 °C, 8 dpi) and R-47 (40.4 °C, 9 dpi), and lasted around five (R-81) to eight days (R-47). For both i-c animals, body temperature remained in the physiologic range throughout the study, with two single exceptions of R-37 (40.0 °C, 10 dpi; 39.6 °C, 25 dpi) (Figure 1A).

At 3 dpi, enlarged cervical lymph nodes (R-47) and mild respiratory signs (R-74) could already be observed, resulting in a clinical score of 1 for one day in both cattle. First, clear clinical signs associated with LSDV infection were observed in R-41, R-81, R-74, and R-47 between 7 and 9 dpi (Figure 1B). Here, reduced activity, decreased feed and water intake in combination with excessive salivation (Figure 2A), enlarged lymph nodes and skin lesions typical of LSDV infections (Figure 2B) were observed. The clinical course of the individual animals R-74 and R-41 became severe quickly, and both animals had to be euthanized at 10 dpi due to animal welfare reasons (Figure 1B). Interestingly, the most severe clinical signs observed were reduced activity and a significant reduction in food and water intake. Although skin lesions were also present, the generalization of skin nodules was not observed. Overall, both animals displayed a systemic clinical course. R-81 and R-47 also displayed a moderate and severe clinical course of lumpy skin disease (LSD) with maximum clinical reaction scores of 6 (R-81, 13 dpi) and 8.5 (R-47, 14 dpi). However, both cattle recovered from the infection, and healing of skin lesions could be observed over time (Figure 2C,D). In contrast, R-84 and R-83 showed only a sub-clinical course of LSD. For R-84, slightly reduced activity could be observed for three days, leading to a clinical score of 1 (11 dpi, 12 dpi) and 0.5 (13 dpi), respectively, whereas for R-83, no clinical signs could be seen at all. Furthermore, both i-c animals did not show any clinical signs (Figure 1B).

Interestingly, the duration of increased body temperature and fever was not directly correlated to the clinical reaction score since R-41 displayed an increased body temperature for only three days but had to be euthanized at 10 dpi, reaching a clinical score of 10. In contrast, R-47 showed an increased body temperature and fever for eight days but recovered completely from the infection.

### 2.2. Viremia and Viral Shedding

EDTA blood and serum samples were used to analyse cell-associated and cell-free viremia, respectively (Figure 3A,B). Additionally, shedding of viral DNA in the nasal and oral fluid was examined using swab samples (Figure 3C,D). Viremia was first detected at 5 dpi in the EDTA blood of three out of six animals experimentally infected with LSDV-V/281-Nigeria (R-84, R-74, R-41) and additionally within the serum of R-41 with Cq-values of around 30–35. In the following days, viremia could be observed in both the EDTA blood and serum samples of almost all animals that were experimentally infected, but with Cq-values still high (between 28.6 (R-84, EDTA blood, 14 dpi) and 37.9 (R-47, serum, 17 dpi)). However, R-83 displayed only cell-free viremia at a single day (17 dpi) with a very high Cq-value of 38.6. In contrast, stronger viremia was observed for both cattle that developed severe LSD and had to be euthanized. Here, lower Cq-values of 27.5 (R-74, EDTA blood), 23.2 (R-41, EDTA blood) and 27.2 (R-41, serum) could be seen. Viremia could not be detected in both i-c animals (Figure 3A,B).

Viral shedding from excretions started at around 7 dpi (R-81, R-41) and 10 dpi (R-47, R-84, R-74, R-83), mainly via nasal fluid. R-83 displayed shedding of viral DNA only sporadically, the other three animals that survived until the end of the study shed the virus over a period of 7 (R-84) to 18 days (R-47) and 21 days (R-81) days with very few exceptions. Cq-values ranged from 27.7 (R-47, nasal swab, 17 dpi) to 38.1 (R-47, oral swab, 17 dpi). Strong nasal shedding of viral DNA could be observed for both cattle that were euthanized at 10 dpi. Here, nasal swab samples displayed a Cq-value of approximately 25, and oral swab samples scored positive with a Cq-value of around 32–35. Both R-74 and R-41 were positive in all four sample matrices on the day of euthanasia. Interestingly, the nasal as well as oral swab samples of both i-c cattle scored positive for capripox viral genome sporadically. The first positive nasal fluid was observed at 10 dpi with Cq-values around 32. At 17 dpi, 21 dpi and 28 dpi, swab samples of R-32 and R-37 were also found to be positive for capripox virus DNA. Here, Cq-values ranged from 33.9 (R-32, nasal swab, 21 dpi) to 36.3 (R-37, nasal swab, 17 dpi). Generally, nasal swab samples turned out to be more sensitive than oral swab samples (Figure 3C,D).

### 2.3. Viral Genome Load in Selected Organ Samples

During the necropsy, samples of cervical lymph nodes and lungs were taken from all experimental animals since these organs both turned out to be most sensitive in previous studies [13]. Furthermore, samples of skin lesions of different stages were collected from R-47. After euthanasia of R-74 and R-41, additional organ samples, as well as samples of skin lesions, were taken. The cervical lymph nodes of all animals that survived until the end of the study were negative using the pan capripox real-time qPCR, and only two animals displayed viral genomes in the lung with a Cq-value of 33.4 (R-84, R-83). A high viral genome load was detected in all organ samples taken from R-74 and R-41 that were euthanized at 10 dpi, verifying the severe generalized infection with LSDV. Cq-values were low in cervical and mediastinal lymph nodes, lung and skin samples. In general, Cq-values between 16 (R-74, cervical lymph node; R-41, mediastinal lymph node, skin samples) and 30 (R-74, liver, spleen; R-41, mesenterial lymph node) could be detected. A comparison of the healing skin lesions and crusts revealed a high viral genome load in both; however, Cq-values of crusts were generally lower, indicating higher loads of viral genome (Table 1).

### 2.4. Serological Response

For serological analysis, the double antigen DA ELISA and the SNT were performed. The first positive result in the ELISA could be detected at 17 dpi (R-84, S/P% ratio of 52), and from 21 dpi onwards, the DA ELISA showed positive results for all four experimentally infected animals that survived until the end of the study (Figure 4A). The first positive SNT results were observed at 21 dpi (R-81 and R-84, ND_50_/mL of 57; R-83, ND_50_/mL of 14), and SNT of R-47 turned positive at 28 dpi (ND_50_/mL of 57). At 28 dpi, ND50/mL between 36 (R-83) and 71 (R-81) could be seen (Figure 4B). For both animals that were euthanized at 10 dpi, no serological response could be observed, which is likely due to the early time point of euthanasia. In addition, both i-c cattle did not show any serological response in both assays during the whole study (Figure 4).

### 2.5. Phylogenetic Analysis of LSDV-V/281-Nigeria

During the phylogenetic analyses of LSDV isolates, clear differences between virulent field strains and attenuated vaccine strains can be observed. In the performed phylogenetic analysis of the present study, LSDV-V/281-Nigeria clusters together with virulent field strains of LSDV (Figure 5). The highest similarity (>99.9%) was observed for the South African strain LSDV NW-LW isolate Neethling Warmbath LW (AF409137, 99.91%) as well as the Balkan isolates LSDV isolate SERBIA/Bujanovac/2016 (KY702007, 99.92%) and LSDV strain Kubash/KAZ/16 (MN642592, 99.94%). The results of the phylogenetic analysis support the results of the in vivo study, classifying LSDV-V/281-Nigeria as a virulent field strain.

## 3. Discussion

In general, the experimental inoculation of cattle with the LSDV V/281-Nigeria field strain resulted in a typical clinical course. The clinic, viremia, viral excretion and serological response were comparable to experimental studies with other virulent LSDV field strains [5,13,14,15,16,19,39,40].

During the presented study, all six cattle that were inoculated with LSDV-V/281-Nigeria showed clinical signs with different disease severity. Two animals (R-84, R-83) displayed a sub-clinical course of LSD. In contrast, the other four cattle were either moderately (R-81) or severely affected (R-47, R-74, R-41), whilst two of the latter animals reached the critical endpoint and had to be euthanized at 10 dpi (R-74, R-41) (Figure 1). This overall pattern is comparable to our experiences with the LSDV-Macedonia2016 field strain after experimental infection of cattle [13,15,40] and with clinical observations after experimental infection with different virulent LSDV isolates described by other groups [5,39,41]. However, compared to our previous studies with LSDV-Macedonia2016, cattle inoculated with LSDV-V/281-Nigeria displayed a more systemic clinical disease with higher clinical scores in categories “general condition” and “food and water intake” instead of high scores for generalized skin lesions and respiratory signs, as were observed for animals infected with LSDV-Macedonia2016 [13,15,40].

Viremia was detected in the EDTA-blood and serum samples of all inoculated cattle at different days between 5 dpi and 17 dpi (Figure 3A,B), which is in line with previous reports [13,14,15,16,40]. Viral shedding was analyzed via nasal and oral swab samples taken during the study. Here, the first positive samples could be observed at 7 dpi, and some animals still displayed viral DNA shedding via nasal and oral fluid at the end of the study (28 dpi) (Figure 3C,D), which is similar but slightly longer than observed for LSDV-Macedonia2016 infections in experimentally challenged cattle [13,15,40]. The first positive results in the ELISA and the SNT could be observed at 17 dpi and 21 dpi, respectively, in all animals that were inoculated with LSDV-V/281-Nigeria that survived until the end of the study (Figure 4). As observed for the other parameters analyzed, this is similar to previous reports describing the onset of an antibody reaction at around 14 to 28 days post-inoculation with virulent LSDV strains [13,15,16,40].

It is known that animals can get infected via virus-containing feed, water [9,12] and likely via licking surfaces recently contaminated by saliva and nasal discharge of infected animals. As licking and sniffing each other is a part of the social behavior of cattle, it is also likely that the infectious virus can be achieved directly from infected cattle [42].

Throughout the experiment, the i-c animals were kept in direct contact with the experimentally infected cattle. Out of the six infected cattle, three showed severe clinical signs of LSD, one was only mildly infected and two did not show any clinical signs. The highest volumes of LSDV in nasal discharge were shed by those animals that were euthanized at 10 dpi. The nasal swabs of the remaining infected cattle remained positive until the end of the experiment but with lower viral loads. The oral swabs of both i-c cattle (R-32 and R-37) tested positive only at 21 dpi. I-c cattle R-37 showed a short fever peak of 40.0 °C at 10 dpi.

Interestingly, the nasal swabs collected from i-c animal R-32 tested positive in real-time PCR, starting at 10 dpi. During the following seven days, the nasal samples of R-32 were negative, but then the animal started to shed the virus again from 17 dpi until 28 dpi when the experiment was finished (Figure 3C,D). The Cq-values of the nasal samples were similar (between 32 and 36.3) to those taken from the experimentally infected animals. As the trial was carried out in vector-free animal facilities, it is clear that R-32 obtained the virus from the experimentally infected cattle via a direct or indirect route around 10 dpi when the highest viral loads were detected in their saliva and nasal discharge. After an incubation period of seven days, R-32 started to shed the virus in their nasal discharge. Neither R-32 nor R-37 showed clinical signs of LSD or viremia, and no viral genome was detected in the organ samples. In addition, neither i-c animal seroconverted.

According to the previously published challenge models [13,43], in order to demonstrate a clinical disease in infected animals, the number of naïve animals exposed to infection needs to be at least six. The number of fully susceptible i-c animals in this study was not sufficient to exclude the possibility that if susceptible animals are kept in direct contact with infected ones, they may become infected with or without showing clinical signs and may further transmit the disease. Previous studies have shown that the incubation time is considerably longer (approximately three to four weeks) when naïve cattle are exposed to lower infectious viral loads, such as when the transmission occurs by vectors or by direct or indirect contact in a natural settings [13,42,44]. Unfortunately, a total experiment duration of 28 days did not allow long enough monitoring of the potential development of viremia and seroconversion in i-c animals.

The transmission of LSDV is believed to occur mainly via blood-feeding insects and ticks [5,6,7,8,45], and direct contact does not play a significant role in the transmission of LSDV [5,41]. The results obtained from our study can also support an alternative possibility: that the positivity of swab samples was an indication for a local, non-systemic replication of LSDV in the nasal and oral mucosa of both in-contact cattle. Although the results of this study may provide further evidence on the direct and indirect route of transmission, more data are required to investigate if the virus can multiply only locally on the oral and nasal mucous membranes of those animals and if a local infection can lead to the further transmission of LSDV to naïve cattle. The findings of this study also underline the value of testing nasal and saliva swabs of suspected cattle during an outbreak investigation or of those animals intended for trade, benefitting the disease control and eradication policies.

In summary, clinical signs induced by this LSDV strain are typical for LSDV infections. However, systemic clinical signs were more prevalent than skin lesions. Overall, all data obtained from the animal study, as well as the phylogenetic analysis, lead to the conclusion that LSDV-V/281-Nigeria is a virulent field strain and is able to induce generalized LSDV infection in cattle.

## 4. Materials and Methods

### 4.1. Animals

Eight female Holstein Friesian cattle between 4–6 months of age were housed in the facilities of the Friedrich-Loeffler-Institut—Insel Riems under biosafety level 4 (animal) conditions. The animals were clinically examined to be healthy, and no acute infections with other pathogens were observed. All respective animal protocols were reviewed by a state ethics commission and have been approved by the competent authority (State Office for Agriculture, Food Safety and Fisheries of Mecklenburg-Vorpommern, Rostock, Germany; Ref. No. LALLF M-V/TSD/7221.3-2-004/18; approval date: 15 March 2018).

### 4.2. Experimental Design and Sample Collection

LSDV-V/281-Nigeria was isolated from a skin sample of cattle taken in May 2018 in Vom, Plateau State, Nigeria (VSD212) during an LSD outbreak and propagated for five passages on Madin-Darby bovine kidney (MDBK) cells. Harvesting of the virus from the cell culture was performed by freezing the infected cell suspension at −80 °C to release the virus from inside the cells. The harvested LSDV was stored at −80 °C until the start of the animal trial. Back-titration of inoculation material on MDBK cells revealed a virus titer of 10^6.4^ cell culture infectious dose_50_ (CCID_50_)/mL. Six of the eight cattle were inoculated intravenously with 4 mL of LSDV-V/281-Nigeria. Two cattle were housed as in-contact (i-c) animals. Rectal body temperature was measured daily from 0 days post-infection (dpi) until 28 dpi. Increased body temperature was defined, ranging from 39.6 °C to 39.9 °C, and fever was defined as >40.0 °C. Furthermore, a clinical reaction score [13] was determined during the same time period. At defined time points of the animal experiment (0 dpi, 3 dpi, 5 dpi, 7 dpi, 10 dpi, 12 dpi, 14 dpi, 17 dpi, 21 dpi and 28 dpi), EDTA blood for analysis of cell-associated viremia, serum samples for examination of cell-free viremia and nasal, as well as oral cotton swab samples for detection of viral shedding, were taken. Serum samples were additionally analyzed serologically. During necropsy, a panel of defined organ samples was obtained: cervical lymph node, mediastinal lymph node, mesenterial lymph node, liver, spleen, and lung, as well as skin lesions of different parts of the body.

### 4.3. Molecular Diagnostics

Organ samples were homogenized in a serum-free medium using the TissueLyser II tissue homogenizer (Qiagen, Hilden, Germany). Genome extraction of samples taken during the animal trial and homogenized tissue samples was performed utilizing the KingFisher Flex System (Thermo Scientific, Darmstadt, Germany) using the NucleoMag Vet kit (Macherey-Nagel, Düren, Germany) according to the manufacturer’s instructions with volume modifications as described previously [46]. To control successful DNA extraction and inhibition-free amplification, an internal control DNA (IC2-DNA) was added to the samples during the extraction process [47,48]. Analyses of the presence of the capripox virus genome were performed using the pan capripox real-time qPCR of Bowden et al. [49] with a modified probe [50], utilizing the PerfeCTa qPCR ToughMix (Quanta BioSciences, Gaithersburg, MD, USA) on a CFX384 Touch Real-Time PCR Detection System (Bio-Rad Laboratories, Hercules, CA, USA).

### 4.4. Serological Examination

Serological examination was performed using two different assays: a commercially available double antigen ELISA from Innovative Diagnostics (ID, Montpellier, France) and the serum neutralization test (SNT). The ID Screen Capripox Double Antigen ELISA (ID) was performed according to the manufacturer’s instructions. For the SNT, serum samples were heat-inactivated for 60 min at 56 °C. Afterwards, log_2_ dilution steps starting from 1:10 dilution were prepared in a serum-free medium in triplicates in a 96 well plate format. Then, 50 µL of LSDV-Neethling vaccine strain [15] were added to each well. After an incubation period of 2 h at 37 °C and 5% CO_2_, 100 µL MDBK cells (approximately 30,000 cells) were seeded into the wells containing the inactivated serum–virus mixture. Following 7 days of incubation at 37 °C and 5% CO_2_, the development of the cytopathic effect was analyzed using the Nikon Eclipse TS-100 light microscope (Nikon, Amsterdam, The Netherlands). The determination of the neutralizing titer was performed using the method of Spearman and Kärber [51,52].

### 4.5. Full-Length Genome Sequencing

The genomic DNA of the cell culture propagated LSDV-V/281-Nigeria was isolated using the MasterPure Complete DNA and RNA Purification Kit (Lucigen/Biozym Scientific GmbH, Hessisch Oldendorf, Germany) according to the manufacturer’s instructions to prepare the sample for sequencing on the Illumina HiSeq 2500 (Illumina, San Diego, CA, USA) platform. Subsequently, the sample was shipped to the ISO17025 accredited Eurofins Genomics lab (Eurofins Genomics Germany GmbH, Ebersberg, Germany) for high-throughput Illumina sequencing (Illumina). The sample was prepared for sequencing according to the company’s workflow. In total, more than 13 million paired reads were produced for further analyses.

Consensus sequences were generated with the mapping tool implemented in Geneious v.11.1.5 (Biomatters, New Zealand) using different LSDV reference sequences (NC_003027, MH893760, KX683219). The respective full-genome sequence obtained was submitted to GenBank under the accession OK318001. For phylogenetic analyses, sequences were aligned using MAFFT, followed by maximum likelihood analyses using Geneious Tree Builder’s Neighbor-Joining method.

## Figures and Tables

**Figure 1 pathogens-11-00016-f001:**
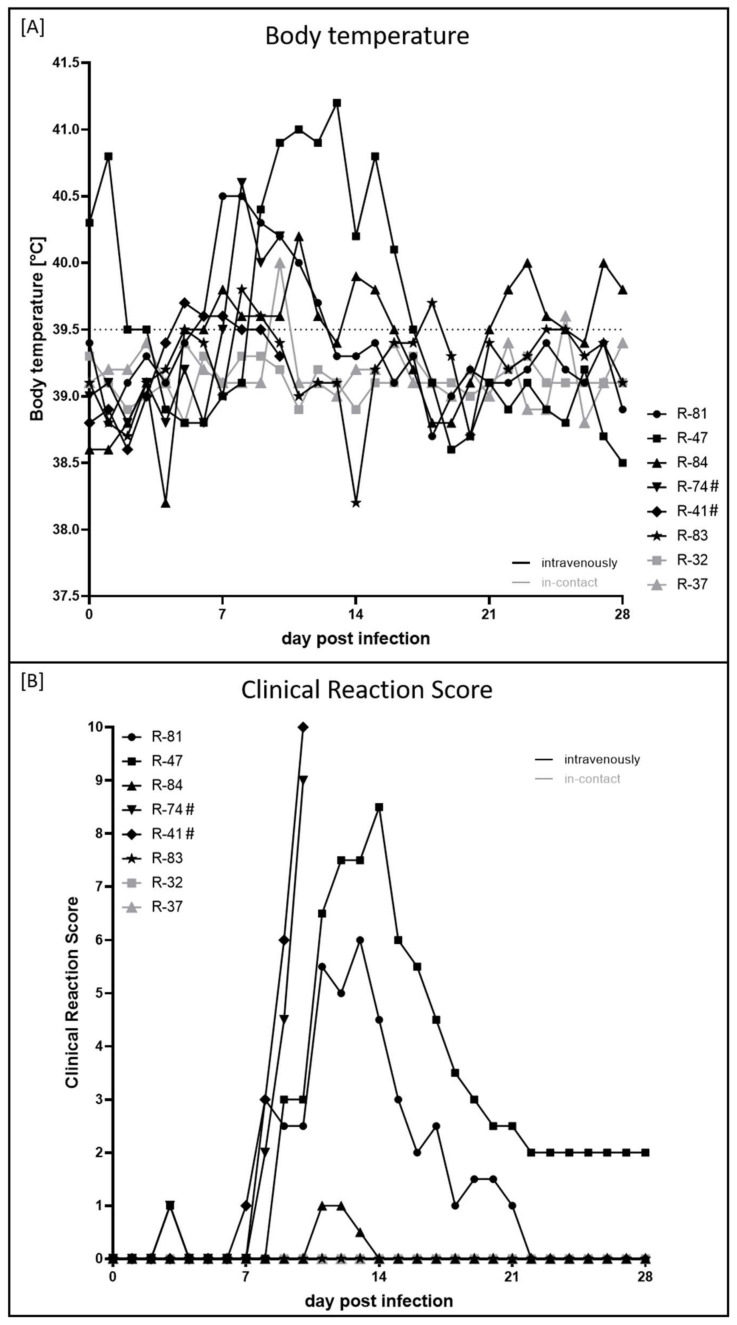
Body temperature and clinical reaction score of cattle experimentally infected with LSDV-V/281-Nigeria. (**A**) Body temperature was measured daily from 0 dpi until 28 dpi. Body temperature ≥ 39.6 °C was defined as increased body temperature, and fever was defined to start at a body temperature ≥ 40.0 °C. (**B**) Clinical reaction score was determined daily during the animal trial. Human endpoint was set at a clinical score ≥ 10 or reaching criteria “abandonment”. Black curves display experimentally infected animals, i-c animals are shown in gray. # marks animals that were euthanized before the end of the study.

**Figure 2 pathogens-11-00016-f002:**
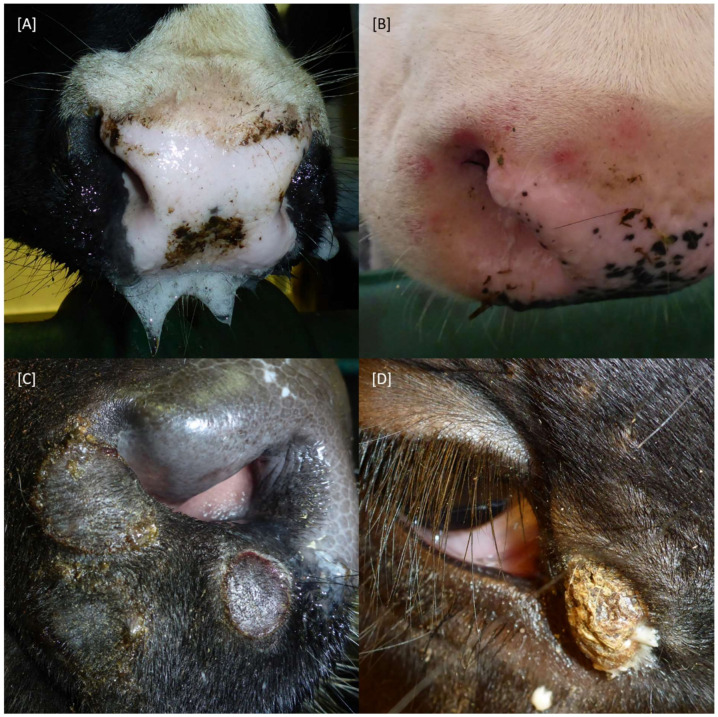
Clinical signs observed after experimental infection of cattle with LSDV-V/281-Nigeria. (**A**) Decreased feed intake was combined with excessive salivation in some animals. (**B**–**D**) Some affected cattle developed characteristic skin alterations. (**B**) Skin lesions still developing. (**C**) Skin lesions during the process of healing. (**D**) Crusts from healing skin lesions on the day of necropsy.

**Figure 3 pathogens-11-00016-f003:**
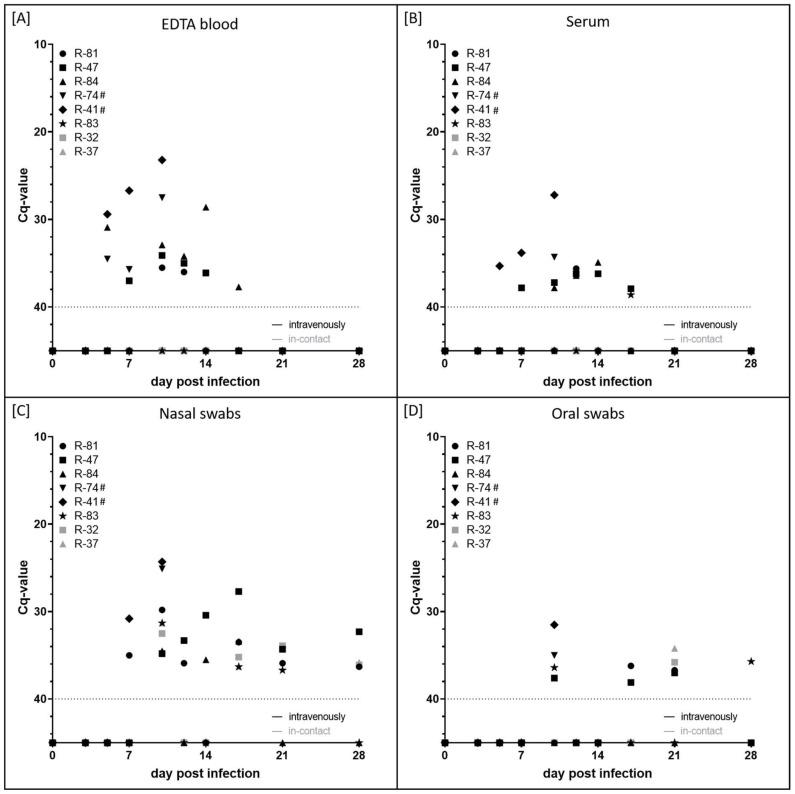
Viremia and viral shedding of animals experimentally infected with LSDV-V/281-Nigeria. (**A**) Cell-associated viremia and (**B**) cell-free viremia were analyzed. In addition, shedding of viral DNA in (**C**) nasal fluid and (**D**) saliva was examined. Cattle experimentally infected are shown in black, gray marks display i-c animals. # displays cattle that were euthanized before the end of the study.

**Figure 4 pathogens-11-00016-f004:**
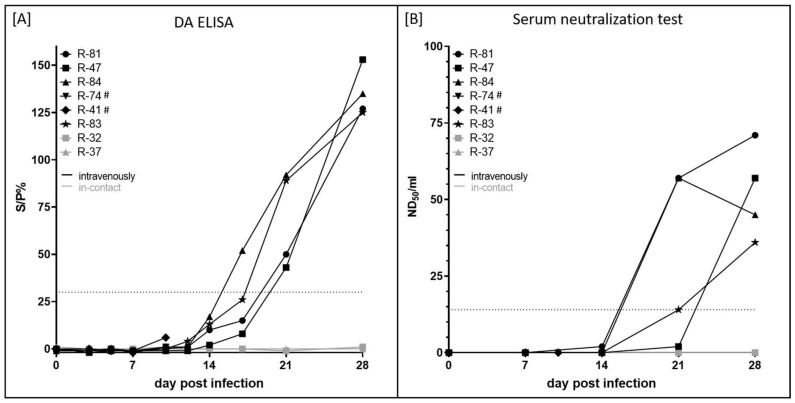
Serological examination of cattle experimentally infected with LSDV-V/281-Nigeria. For serological analysis, (**A**) the DA ELISA and (**B**) the SNT were used. Due to the manufacturer, the DA ELISA is positive with an S/P% ratio ≥ 30. The SNT was defined as positive with ND_50_/mL ≥ 14. # displays animals that were euthanized before the end of the study.

**Figure 5 pathogens-11-00016-f005:**
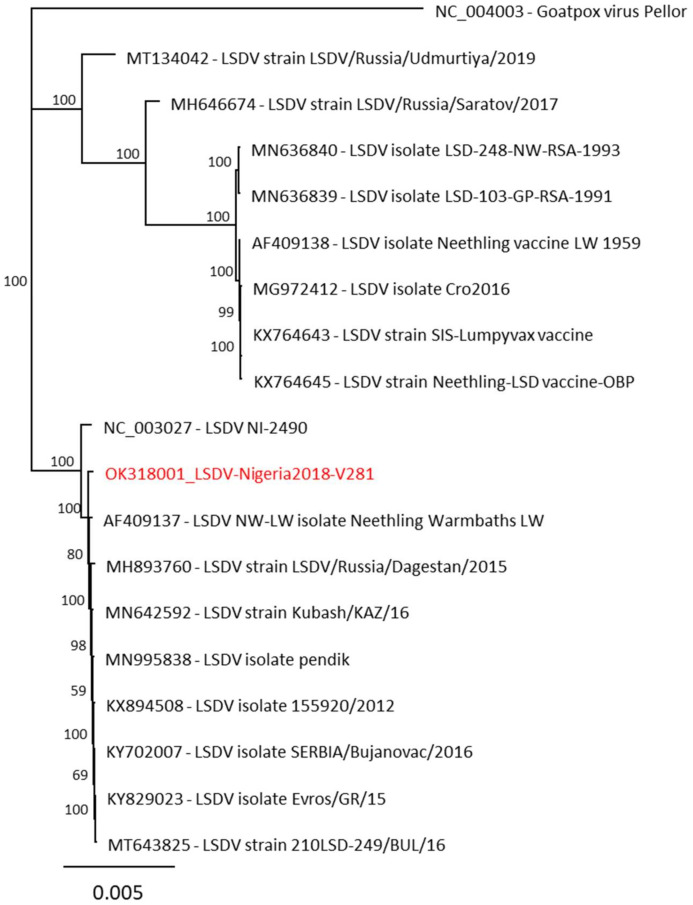
Phylogenetic relationship of LSDV-V/281-Nigeria isolates (red). Illumina sequencing was performed to obtain the full-length genome sequence.

**Table 1 pathogens-11-00016-t001:** Real-time PCR results (Cq-values) of different organ samples taken from cattle experimentally infected with LSDV-V/281-Nigeria during necropsy (no data display samples not taken).

	Experimentally Infected Animals						i-c Animals	
Sample Material	R-81	R-47	R-84	R-74	R-41	R-83	R-32	R-37
Cervical lymph node	no Cq	no Cq	no Cq	15.9	21.8	no Cq	no Cq	no Cq
Mediastinal lymph node				22.7	16.0			
Mesenterial lymph node				37.3	28.3			
Liver				29.5	21.8			
Spleen				28.9	19.2			
Lung	no Cq	no Cq	33.4	26.1	23.2	33.4	no Cq	no Cq
**Location of Skin Sample**								
Anus				25.3	17.8			
				24.6	16.3			
				24.2	18.5			
Muzzle				22.4				
Subcutis		no Cq						
		22.0						
		23.6						
Lung		31.8						
		28.3						
		27.1						
		28.8						
		28.4						
Healed skin lesion (see Figure 2C)		28.2						
		26.5						
		28.5						
		27.0						
		34.0						
Crust of skin lesion (see Figure 2D)		19.7						
		20.2						
		21.7						
		18.3						
		19.0						
		18.1						
		17.6						
		21.4						
		20.7						

## Data Availability

The data of this study are available in the presented manuscript.

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
