# Peer review of "Characterization of a Nigerian Lumpy Skin Disease Virus Isolate after Experimental Infection of Cattle"

_pathogens, 2021, doi:10.3390/pathogens11010016_

Round 1

Reviewer 1 Report

In this manuscript entitled “Characterization of a Nigerian lumpy skin disease virus isolate after experimental infection of cattle”, the authors described the experimental infection of six naïve cattle with a low passaged LSDV isolated from an outbreak in Nigeria in 2018. Two in-contact animals were kept along with the experimentally infected animals, whilst clinical, molecular and serological data were collected during the 28 day trail. Based on the data obtained, the authors concluded that the aforementioned isolate form Nigeria in 2018 was highly pathogenic in naïve cattle.

The manuscript is comprehensively constructed and described the aim, materials and methods used as well as the results obtained in a clear and logical manner. The conclusions reached based on the data presented are sound and the manuscript is recommended for publication. There are some minor suggestions listed below.

Minor suggestions:

  • Line 16. Omit “most”.
  • Line 22. Omit “few”
  • Lines 33 to 34: “lumpy skin disease virus”, “sheeppox virus” and “goatpox virus“ should not be in italics.    
  • Line 69: Change “Rough” to “Initial” or “Rudimentary”
  • Line 71: Isolate NI-2940 originate from Kenya in 1958, not South Africa.
  • Lines 76: Change “turned out to be” to “proved”.
  • Line 84: Add “the” between “In” and “case”.
  • Line 97: add “the” before “most”, or consider changing the sentence.
  • Line 169: Omit “very”
  • Line 179: Include “double antigen” before DA, since it is the first description of the abbreviation.
  • Line 182: Change” The SNT started to give positive results” to “The first positive SNT results were observed”
  • Lines 208 to 211. Consider changing the sentence.
  • Lines 213 to 215: Suggested changes: In contrast, the other four cattle were either moderately or severely affected, whilst two of the latter animals reached the critical endpoint and had to be euthanized…”
  • Line 223: Change “they could be” to “were”
  • Line 255: Change “achieved” to “obtained”
  • Line 306: Change “out of” to “of the”

Reviewer 2 Report

See attachment.

Reviewer 3 Report

I suggest authors to add/clarify if there is a specific reason why the particular isolate was selected to be deeper characterized? Is that because of similarity with strains that caused outbreaks in Balkans or something else? 
